# Under Psychological Safety Climate: The Beneficial Effects of Teacher–Student Conflict

**DOI:** 10.3390/ijerph19159300

**Published:** 2022-07-29

**Authors:** Ruoying Xie, Jinzhang Jiang, Linkai Yue, Lin Ye, Dong An, Yin Liu

**Affiliations:** 1School of Media & Communication, Shanghai Jiao Tong University, Shanghai 200240, China; yuraxie@sjtu.edu.cn (R.X.); ye_lin@sjtu.edu.cn (L.Y.); liuyin0531@sjtu.edu.cn (Y.L.); 2USC-SJTU Institute of Cultural and Creative Industry, Shanghai Jiao Tong University, Shanghai 200240, China; 3School of Art, Southeast University, Nanjing 211189, China; donganphd@126.com

**Keywords:** teacher–student relationship, cognitive conflict, affective conflict, students’ innovative competence, psychological safety climate

## Abstract

Previous studies have mainly focused on the negative effects of teacher–student conflict; the positive effects of conflict have rarely been mentioned. This paper suggests that encouraging conflict could act as a teaching method to improve students’ innovative competence. This study has two objectives: (1) to examine how various types of teacher–student conflict affects students’ innovative competence and (2) to identify the mediating role of a psychological safety climate in the association between conflict and students’ innovative competence. To achieve the objectives, we used evidence from 1207 university students. Multivariable logistic regression analysis revealed that conflicts were associated with students’ innovative competence, and the mediation role of a psychological safety climate is significant. Specifically, the results revealed that Cognitive Conflict had significant positive effects on students’ innovative competence, whereas Affective Conflict had a significant negative effect on students’ innovative competence. In addition, we clarified a psychological safety climate as the boundary condition for the relationship between conflict and students’ innovative competence.

## 1. Introduction

Students’ innovation competence has been identified as an important learning goal at the university level [1,2]. How to improve students’ innovative competence is emphasized and debated in university pedagogy [2,3]. Innovation is “the intentional introduction and execution within a group or organization of ideas, processes, products or procedures, new to the relevant unit of adoption, designed to significantly benefit the individual, the group, the organization or wider society” [1]. The embodiment of students’ innovative competence is the new viewpoint [1,2].

Conflict is a common and inevitable occurrence among teacher–student interactions in a university [4]. Previous studies concluded that conflict is harmful because it causes students’ academic risk, disengagement, withdrawal, and failure at school [5,6,7]. Considering the detrimental consequences of conflict, researchers have developed a series of management methods to avoid or suppress it [8,9]. However, in team studies, it has been proven that cognitive conflict is conducive to enterprise performance and employees’ creativity [10,11,12]. Previous studies have identified a variety of predictors of students’ innovative competence, such as student–teacher trust and the student–teacher relationship [5,6,13,14,15]. However, little research has considered that conflict is conducive to innovative competence. Could we suppose that teacher–student conflict is a way to stimulate students’ innovative competence?

Conflict is defined as a difference of opinion between two or more people or groups [16,17,18]. As prior research mentioned, conflict can be divided into two forms: cognitive conflict and affective conflict [10,19,20,21,22]. During the teacher–student interaction, cognitive conflict refers to disagreement in viewpoints, ideas, and opinions during the teacher–student interaction [23,24]. Affective conflict is defined as disagreements related to interpersonal emotional incompatibilities induced by differing personalities and values [25,26]. For the sake of an orderly, disciplined teaching environment, positive teacher–student relationships were recommended, and students were encouraged to comply with school rules; conflict has always been considered disruptive and aggressive behavior [26,27].

However, in the case of knowledge innovation, this overemphasis on negative consequences may detract attention from the beneficial effects that cognitive conflict may have. Through cognitive conflict, students can debate openly and voice dissenting views while interacting with teachers. It is likely to promote divergent thinking and the sharing of a broader range of thoughts, which can benefit students’ innovative competence. Hence, this paper attempts to reveal the beneficial influence of cognitive conflict and proposes an innovative teaching method.

In addition, the key to using conflict to improve innovative competence is enhancing students’ willingness to express their objections [28,29]. Therefore, a positive psychological safety climate between teachers and students is essential. Psychological safety is an individual’s shared belief about whether it is secure to engage in risk-taking [30,31]. Students studying in a psychologically safe environment feel a sense of openness, which allows them to question the teacher without worries and contribute more diverse ideas [32,33].

According to the above, we found that the types of teacher–student conflicts remain ambiguous. Moreover, there is a paucity of empirical evidence governing the positive effects of teacher–student conflict. Previous literature has emphasized the importance of a psychological safety climate in the classroom [32,34], but the role of a psychological safety climate in the relationship between conflict and innovative competence has not been investigated. Hence, this study has three objectives: (1) describe the process of two kinds of teacher–student conflicts and explain the features and different situations during teacher–student interaction; (2) examine how various types of teacher–student conflicts affect students’ innovative competence; (3) identify the mediating role of a psychological safety climate in the association between conflict and students’ innovative competence.

### 1.1. Conflict between Teacher and Student

Differences in attitudes, beliefs, values, or needs induce conflict [16,17,18]. Previous studies have demonstrated the obvious and observable characteristics of conflict, such as angry words, oppositional behavior, or fierce objection [7,33]. However, these studies mentioned only one type of conflict: affective conflict [7,25,33]. Compared to affective conflict, cognitive conflict has received less attention.

Originating from team management, conflict studies might transform its connotation in the classroom environment. Therefore, it is necessary to clarify the conflict concept in teacher–student interaction situations. In team studies, cognitive conflict occurs when team members debate different views about a task [10,12,21]. Affective conflict involves disagreements of a personal nature, such as power struggles or personal incompatibilities [10,12,21].

Cognitive conflict happens during the teacher–student interaction when there are arguments over knowledge differences in viewpoints [21,23]. Conflict is an unavoidable occurrence [24] because teachers and students play distinct roles in the classroom, and their knowledge backgrounds, experiences, and priorities are different [35]. Their understanding of class content may be subversive, and students may not always completely understand the theory and knowledge, nor do they always fully accept all of the teachers’ opinions.

On the other hand, affective conflict arises from interpersonal incompatibilities [21]. The incompatibilities could be differences in personality, values, personal taste, political preferences, or interpersonal styles [36]. Affective conflict is often accompanied by intense behavior and negative emotions [25]. For example, teachers may arbitrarily interrupt students’ speech, ignore their questions, and even criticize them because of students’ contrary ideas [37]. In these situations, affective conflict may occur between the student and teacher.

### 1.2. Psychological Safety Climate

A psychologically safe climate is characterized as open, authentic, and direct [30,38]. “Open” refers to an individual addressing concerns and disagreements publicly in the organization [30,38]. “Authentic” is defined as people transparently expressing and behaving according to their personal values without others’ judgment [30,38]. “Direct” is explained as team members not needing to choose their words overly cautiously before starting a conversation [31,39].

A psychological safety climate was suggested to significantly positively affect students’ motivation, engagement, long-term memory, and academic achievement [40]. A positive psychological safety climate is student-centered, with warmth, respect, and responsiveness [41]. In contrast, a negative psychological safety climate is related to high levels of anger and sarcasm [42]. A psychological safety climate helps students to take risks with a low concern about embarrassment, rejection, or punishment from teachers [43]. It allows students to focus on their learning rather than being distracted by worries of being ridiculed [44].

Evidence demonstrated that psychological safety was a context-shifting state, which could have a buffering effect that supported learning in high-stress situations [39]. In addition, a psychological safety climate has been proven to be correlated with better academic achievement and less disruptive behaviors [34]. Hence, we reasonably expect that a psychological safety climate can amplify the advantage of conflicts. Therefore, this paper will explore the mediating role of a psychological safety climate in conflict situations.

#### 1.2.1. The Relationship between Cognitive Conflict and Students’ Innovative Competence

Cognitive conflict occurs when the teacher and students discuss various preferences and opinions about a knowledge issue [11,23]. The constructive debate and exchange of perspectives may help students to understand knowledge from different angles and evaluate other solutions to problems. Cognitive conflict also provides a chance for students to receive, justify, evaluate, and refine their ideas from teachers. Asking teachers questions can enhance students’ learning initiative and improve students’ class involvement [6,45]. Even if these questions are not helpful or disruptive, the behavior of putting forward questions is valuable [46] because it is an outcome of students’ deep thought, which can also enhance students’ innovative competence. Rather than the students’ excessive conformity with teachers, cognitive conflict may improve the judgments and interpretation of the knowledge.

In sum, cognitive conflict promotes divergent perspectives, which benefit students’ innovative competence. Accordingly, we propose:

**Hypothesis** **1** **(H1).**
*During teacher–student interaction, cognitive conflict is positively correlated with students’ innovative competence.*


#### 1.2.2. The Relationship between Affective Conflict and Students’ Innovative Competence

Affective conflict is often accompanied by negative attitudes and emotions; it creates tensions between teachers and students, as well as anxiety and unpleasant feelings [21,36]. These negative emotions may cause students’ misbehavior and problematic behavior, such as students’ aggression toward teachers [36,47]. It reduces classroom efficiency and even induces skipping classes and dropping out of school [37,47,48].

Affective conflict also reduces teacher–student trust [37,48]. It decreases communication frequency, which inhibits student–teacher interaction and reduces knowledge sharing. During affective conflict, even though a student is interested in the topic of discussion, they choose to avoid it or do not respond to teachers. As a result, the misunderstandings deepen between students and teachers, which causes conflict escalations [49].

Based on the discussion, we propose:

**Hypothesis** **2** **(H2).**
*During teacher–student interaction, affective conflict is negatively correlated with students’ innovative competence.*


#### 1.2.3. The Relationship between Psychological Safety Climate and Students’ Innovative Competence

A psychological safety climate affects how people perceive conflict [29,47]. It may reduce the possible risks and hazards of cognitive conflict. A psychological safety climate relieves pressure on students fearing negative consequences, for example, being ignored or punished by teachers and feeling embarrassed in the classroom [32]. In a psychological safety climate, students do not have to be concerned about self-image and saving face [32]. Hence, we suppose that a psychological safety climate helps students to engage in conflict or confrontation without worries. As a result, this may help students to better express different or risky ideas in the case of cognitive conflict.

In addition, a psychological safety climate can reassure students that conflict does not include personal judgment but only relates to knowledge issues [47]. In a psychological safety climate, teachers respect students’ suggestions and encourage them to raise objections, which provides an open communication environment [34]. A psychological safety climate increases teacher–student trust, and both students and teachers increase their willingness to communicate and express their true thoughts [39]. In this way, misunderstandings between students and teachers can be eliminated. A psychological safety climate may weaken or solve affective conflict, which also enhances students’ innovative competence.

In the above cases, we propose the following hypotheses:

**Hypothesis** **3** **(H3).**
*A psychological safety climate mediates the positive relationship between teacher–student cognitive conflict and students’ innovative competence.*


**Hypothesis** **4** **(H4).**
*A psychological safety climate mediates the negative relationship between teacher–student affective conflict and students’ innovative competence.*


In order to achieve objective 2 (examine how various types of teacher–student conflicts affect students’ innovative competence), H1 and H2 were demonstrated based on conflict theory. For objective 3 (identify the mediating role of psychological safety climate in the association between conflict and students’ innovative competence), H3 and H4 were demonstrated based on the previous literature about psychological safety climate. Based on the discussion, the theoretical model and hypotheses are shown below (Figure 1).

## 2. Materials and Methods

### 2.1. Data Collection

The following two methods were used to assess Hypotheses 1–4: Questionnaire survey and In-depth interviews. The flow chart is shown in Figure 2.

To begin, we created a questionnaire along with classic scales to assess Hypotheses 1–4 (details on measurements).

Secondly, as a preliminary step in the research, we disseminated a pilot survey to evaluate the research’s effectiveness.

Thirdly, the data from the pilot survey were analyzed using the Structural Equation Model. The effective items (Cronbach’s alpha > 0.80) were maintained, while the lesser validation items (Cronbach’s alpha < 0.30) were eliminated. In-depth interviews were performed to refine the left items (0.30 < Cronbach’s alpha < 0.80), resulting in a higher Cronbach’s alpha (over 0.80) (All of the elements are effective; check Table 1 for further information).

For example, pilot tests of the survey revealed that 91.8% of students chose “strongly agree” or “agree” in response to the item “I will make friends with my teacher”. However, the in-depth interviews indicated that students disliked their teacher and would not recommend the course to other classmates (79.1%). We found that the reason was mainly that students thought their teacher would read their responses to the survey and that they were related to their grades in the course. Hence, we added an explanation in the instructions to clarify that all identifying information was hidden, and teachers and classmates did not have the right to read it. In this way, students could respond to the questionnaire truthfully, and the interpretive validity of the survey was increased.

Finally, the pretesting questions were refined or adapted in the final questionnaire, and we distributed it again. Teachers and volunteers administered the survey, with 22.4% of the students filling out a paper version of the survey in their classrooms and 87.6% completing an electronic version of the questionnaire via e-mail. As a result, the final data set consists of 1027 entries (response rate = 92.9%). The students’ ages ranged from 17 to 24 years, and 53.9% of the 1027 students were female. Confidentiality and anonymity were ensured, and all participation was voluntary and anonymous. After submitting the assessment, each student was provided a gift as a participation incentive.

### 2.2. Sample Description

The sample was obtained non-randomly and through convenience sampling in this study. The final sample consisted of 1183 university students from 7 American universities who were majoring in a variety of subjects (Arts, Science, Business, Engineering and Technology, Literature, Language, and Social Science). A total of 78 questionnaires were discarded because the students did not complete the survey or completed it in less than 8 min with inadequate information. As a consequence, 1105 surveys were completed and returned.

The following were the questionnaire’s inclusion criteria:
Students are full-time undergraduates from 1st to 5th grade;All of the students are over 18 years old;All the students have the experience of offline class (the object of this study did not refer to virtual teacher–student interaction).

For In-Depth Interviews, the following criteria were used: Students who volunteered for the interviews (at the end of the questionnaire, there was a suggestion to “Leave your e-mail address if you want to be interviewed”).

### 2.3. Data Process

Firstly, the reliability of the questionnaire was analyzed (Cronbach’s alpha, Composite reliability, Mean variance extracted and Variance Inflation Factor (Table 1).

Secondly, a descriptive analysis of the obtained results was carried out to report the means, standard deviations, and inter-scale correlations between all of the study variables (Table 2). The results of the correlation analysis revealed that all the variables had significant associations, which supported the use of regression and mediating analysis in the following steps.

Subsequently, Hypotheses 1–4 are supported by the Structural Equation Model (SPSS 26).

Finally, to further test the mediating effect (H3, H4), we estimated the indirect, direct, and total effects, as well as their 95% bias-corrected confidence intervals (CI); the 5000-replication bootstrapping procedure was used based on the PROCESS macro adapted from Hayes. H3 and H4 were also supported.

### 2.4. Measures

To measure students’ innovation competence, we adapted the scales by Ovbiagbonhia [1]. It comprises three different types of competencies: leadership, solving ambiguous problems, and risk tolerance. For example, the students were asked to reply to the item, “I am excited by unanswered questions”.

Cognitive conflict was assessed using the scale developed by Mooney [21]; a sample item is, “Teacher and I frequently have disagreements about the knowledge issue”.

Affective conflict was assessed using the scale adopted by Mooney [21]; a sample item is, “I will not make friends with my teacher”.

Psychological safety climate was assessed using the scale developed by Newman [47]; a sample item is, “It is easy to put forward different ideas in the class”.

## 3. Results

Cognitive conflict was positively correlated with students’ innovative competence (*r* = 0.771, *p* < 0.01). Affective conflict negatively predicted students’ innovative competence (*r* = −0.582, *p* < 0.01). Cognitive conflict had positive effects on PSC (*r* = 0.358, *p* < 0.01), and affective conflict negatively predicted PSC (*r* = −0.454, *p* < 0.01). We also found that a psychological safety climate and students’ creativity were positively correlated (*r* = 0.690, *p* < 0.01).

The results of the correlation analysis revealed that all the variables had significant associations, supporting the use of regression and mediating analysis in the following steps (Table 2).

The results of the hierarchical regression (M4) support Hypotheses 1 and 2, which achieved objective 2 (Table 3). Cognitive conflict had significant positive effects on students’ innovative competence (β = 0.319, *p* < 0.01). Affective conflict was significantly negatively related to students’ innovative competence (β = −0.505, *p* < 0.01).

Cognitive conflict had significant positive effects on PSC (β = 0.275, *p* < 0.01), and affective conflict had a significant negative effect on PSC (β = −0.209, *p* < 0.01) (M2). PSC was significantly and positively correlated with students’ innovative competence (β = 0.227, *p* < 0.01) (M5). These patterns are consistent with Hypotheses 3 and 4, which resulted in verifying the relationship between psychological safety climate and students’ innovative competence (objective 2).

To further test the mediating effect, we estimated the indirect, direct, and total effects, as well as their 95% bias-corrected confidence intervals (CI); the 5000-replication bootstrapping procedure was used based on the PROCESS macro adapted from Hayes [49]. As shown in Table 4, the indirect effects of CC and AC on PSC were significant (indirect effect = 0.052, LLCI = 0.037, ULCI = 0.065; indirect effect = 0.983, LLCI = 0.692, ULCI = 0.767, respectively). However, the direct effects were statistically insignificant (i.e., the 95% CIs included zero) for two types of conflict.

These results thus support the hypothesized mediation model (H3 and H4), indicating that only in a PSC can cognitive conflict promote students’ innovative competence, and only in a PSC can affective conflict be reduced, which also enhances students’ innovative competence.

## 4. Discussion

This study contributes to nascent literature that seeks to understand how teacher–student conflict affects students’ innovative competence and the buffering effect of a psychological safety climate under conflict situations.

Firstly, our findings supported that cognitive conflict promoted students’ innovative competence. The literature proved that cognitive conflict had positive outcomes (such as group performance) in companies and top management teams [16,18,19,20,21,22]. We found that cognitive conflict between teachers and students had a beneficial impact as well. Cognitive conflict facilitates the surfacing of different ideas and viewpoints. It helps students reconsider knowledge from various angles. In contrast to students’ silence or blind acceptance of knowledge, cognitive conflict increases learning feedback and more extensive debates, which results in new understanding and more perspectives.

Secondly, consistent with the studies of negative outcomes of affective conflict [3,7,36,37], the results indicated that affective conflict was negatively correlated with students’ innovative competence. Affective conflict results in discussion that is off-topic for class content, and it evokes negative emotions and amplifies aggression. These findings achieved the first goal mentioned earlier.

Thirdly, we found that a psychological safety climate strengthened the advantages of cognitive conflict and weakened the disadvantages of affective conflict, which proved the third goal. Previous studies have found that a psychological safety climate promotes knowledge sharing, learning behavior, and team creativity [28,30,31]. Our findings are consistent with these conclusions.

A psychological safety climate transforms a tense communication environment in cognitive conflict into an open communication environment. The conversation in cognitive conflict is unusually direct and confrontational, and it is easy to trigger intense behavior or negative emotion [21,23]. However, a psychological safety climate relieves pressure and negative emotion and provides students a secure environment to share embarrassing or contrary ideas. By freeing them from worrying about being ignored or being punished, students do not need to focus on self-protection.

At the same time, a psychological safety climate transforms the tense interpersonal atmosphere into a warm and respectful environment. A psychological safety climate increases the trust between teachers and students, which allows or solves interpersonal contradictions.

## 5. Limitations and Suggestions for Future Research

This study has a few limitations that should be highlighted for future research. The link between cognitive conflict and affective conflict was not considered. Moreover, regarding the measurement, the evaluation of students’ innovative competence is self-evaluation, and the teachers’ reports can be added. For future studies, we recommend exploring the impact of different conflicts on teachers’ behavior and cognition. Researchers can also explore other factors that may help cognitive conflict boost students’ innovative competence.

## 6. Conclusions

These results have important theoretical implications for teacher–student conflict research. Firstly, few studies have systemically assessed two types of teacher–student conflict. The present study investigated how different types of conflict are associated with students’ innovative competence and provided empirical evidence for the positive effects of cognitive conflict on students’ innovative competence. Although a growing body of research has operationalized a positive psychological safety climate as a predictor of academic achievement, effective violence prevention, and students’ healthy development, the mediator role of a psychological safety climate under conflict situations has not been clarified. We explored the psychological safety climate as the boundary condition between conflict and students’ innovative competence. Furthermore, a psychological safety climate reduces the energy needed to regulate affective conflict and to deal with other distracting issues. It assures that cognitive conflict does not include personal judgment but relates only to knowledge issues.

We envision that the current research will assist and guide teachers by clarifying how conflict can improve students’ innovative competence. Considering the positive association between cognitive conflict and students’ innovative competence, teachers should guide and encourage cognitive conflict and use it as an opportunity to cultivate students’ innovative competence. In contrast, due to the detrimental effects of affective conflict, teachers should strive to avoid it and resolve it as early as possible when it arises.

In university, coping with conflict is not only the chief responsibility but also the toughest challenge for teachers. However, our findings imply that conflict could be beneficial when a psychological safety climate is fostered. Thus, in order to capitalize on the benefits of conflict and stimulate students’ innovative competence, teachers should seek to cultivate and maintain the development of a psychological safety climate.

## Figures and Tables

**Figure 1 ijerph-19-09300-f001:**
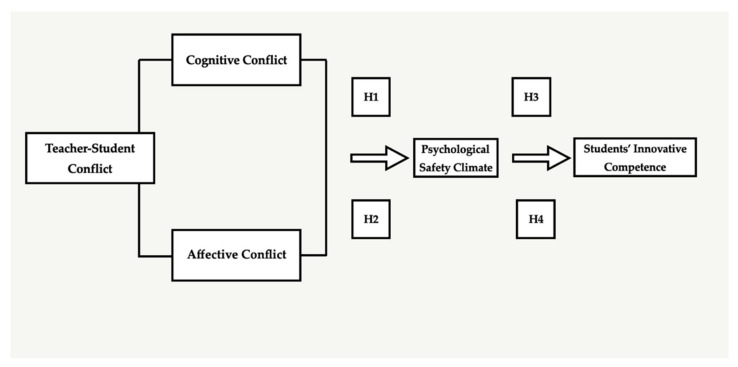
Theoretical Model and Hypotheses.

**Figure 2 ijerph-19-09300-f002:**
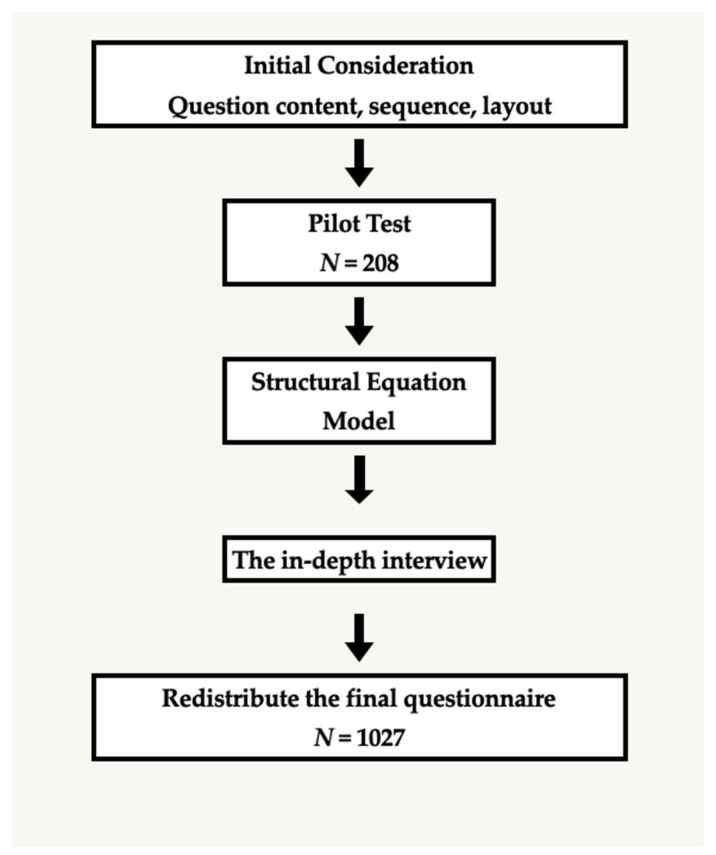
Flow Chart.

**Table 1 ijerph-19-09300-t001:** The validation of measurements.

Construct	Item	Factor Loading	Cronbach’s Alpha	CR	AVE	VIF
Students’ innovation competence	S1	0.683	0.858	0.9	0.77	1.82
	S2	0.783				
	S3	0.784				
	S4	0.801				
	S5	0.734				
	S6	0.699				
	S7	0.733				
	S8	0.775				
	S9	0.814				
	S10	0.812				
	S11	0.807				
	S12	0.698				
Cognitive conflict	C1	0.677	0.801	0.9	0.8	2.03
	C2	0.731				
	C3	0.742				
	C4	0.683				
Affective conflict	A1	0.672	0.829	0.8	0.76	1.98
	A2	0.765				
	A3	0.838				
	A4	0.699				
Psychological safety climate	P1	0.723	0.815	0.8	0.62	2.01
	P2	0.699				
	P3	0.721				
	P4	0.753				

**Table 2 ijerph-19-09300-t002:** Means, standard deviations, and correlations.

Variable	M	SD	1	2	3	4	5	6
1.G1	1.52	0.782	1					
2.G2	2.22	0.838	0.568	1				
3.CC	2.54	0.704	0.576	0.663	1			
4.AC	2.46	0.622	0.323	0.248	0.232	1		
5.PSC	2.78	0.677	0.317 **	0.269 **	0.358 **	−0.454 **	1	
6.SIC	2.97	0.789	0.797	0.590 **	0.771 **	−0.582 **	0.690 **	1

** *p* < 0.01 (two-tailed) G1 = Gender; G2 = Grade; CC = Cognitive conflict; AC = Affective conflict; PSC = Psychological safety climate; SIC = Students’ innovative competence.

**Table 3 ijerph-19-09300-t003:** Hierarchical regression analysis.

Variable		PSC		Students’ Innovative Competence		
		M1	M2	M3	M4	M5
CV	G1	0.049 **	0.020	0.088	0.046	0.048
	G2	0.431 **	0.162 **	0.167	0.124	0.069
IV	CC		0.275 **		0.319 **	0.391
	AC		−0.209 **		−0.505 **	−0.301
MV	PSC					0.227 **
	R2	0.122	0.403	0.161	0.443	0.594 **
	∆R2	0.128	0.281	0.161	0.282	0.151 **
	F	27.868 ***	119.031 ***	26.261 ***	297.451 ***	128.883 ***

** *p* < 0.01; *** *p* < 0.001 (two-tailed) CV = Control variable; IV = Independent variable; MV = Mediating variable; G1 = Gender; G2 = Grade; CC = Cognitive conflict; AC = Affective conflict; PSC = Psychological safety climate; SIC = Students’ innovative competence.

**Table 4 ijerph-19-09300-t004:** Result of mediation models.

				Effect	LLCI	ULCI
Direct effect 1	CC	→	SIC	0.074	−0.039	0.108
Indirect effect 1	CC	→PSC	SIC	0.052	0.037	0.065
Direct effect 2	AC	→	SIC	0.073	−0.041	0.091
Indirect effect 2	AC	→PSC	SIC	0.983	0.692	0.767

G1 = Gender; G2 = Grade; CC = Cognitive conflict; AC = Affective conflict; PSC = Psychological safety climate; SIC = Students’ innovative competence.

## Data Availability

The data presented in this study are available on request from the corresponding author.

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
