# Peer review of "Under Psychological Safety Climate: The Beneficial Effects of Teacher–Student Conflict"

_ijerph, 2022, doi:10.3390/ijerph19159300_

Round 1
Reviewer 1 Report
Dear authors,
I consider that the manuscript presented is interesting; in the first place, it is a contribution to literature, but above all to teachers, because they can understand that, improving the psychological climate with their students.
I have some questions and suggestions:
• What are the attributes of the innovation competence that were measured in the questionnaire?
• The results specify how objectives 2 and 3 were achieved, but it is not clearly identified how objective 1 is met.
• Could you explain in more detail the characteristics of the questionnaire? How many items did you construct? What were the response options?
• I suggest you review the abstract, to better understand the development of the research, should the objectives be written at the end?
I hope these comments help to strengthen the article. Success with the publication.
Best regards.
Author Response
We appreciated your careful reading of our manuscript and the constructive comments. We have thoroughly addressed these comments in the revision, incorporating the majority of recommended changes as described below in our detailed point-by-point responses. The revisions of the manuscript according to the comments are marked in red (Response), and purple (Refine paragraph in the paper), respectively. The detailed revisions in the manuscript and the responses are as follows.
Comment 1: What are the attributes of the innovation competence that were measured in the questionnaire?
Response 1: To measure students’ innovation competence, we adapted the scales by Ovbiagbonhia[1]. It comprises three different types of competencies: leadership, solving ambiguous problems, and risk tolerance.
Comment 2: The results specify how objectives 2 and 3 were achieved, but it is not clearly identified how objective 1 is met.
Response 2: Objective 1 was detected because it was the conception introduction of different conflicts. The objectives only left 2 and 3.
Comment 3: Could you explain in more detail the characteristics of the questionnaire? How many items did you construct? What were the response options?
Response 3: The questionnaire had 30 items with 4 constructs: Students' innovation competence, Cognitive conflict, Affective Conflict, and Psychological safety climate. Each construct was measured by the established and classical scales.
To measure students’ innovation competence, we adapted the scales by Ovbiagbonhia [1]. For example, the students were asked to reply to the item, “I am excited by unanswered questions”.
Cognitive conflict: It was assessed using the scale developed by Mooney [21]; a sample item is “Teacher and I frequently have disagreements about the knowledge issue”.
Affective conflict was assessed using the scale adopted by Mooney [21]; a sample item is “I will not make friends with my teacher”.
Psychological safety climate was assessed using the scale developed by Newman [47]; a sample item is “It is easy to put forward different ideas in the class”.
The reliability of the questionnaire has been proved. The factors (Factor Loading, Cronbach's alpha, CR, AVE, VIF) were shown in Table 3. We used the five-level Likert scale as the response, it is a type of psychometric response scale in which responders specify their level of agreement to a statement typically in five points. All items used the response alternatives of Strongly disagree; Disagree; Neither agree nor disagree; Agree; Strongly agree, coded into 1, 2, 3, 4, and 5, respectively.
Comment 4: I suggest you review the abstract, to better understand the development of the research, should the objectives be written at the end?
Response 4:
The abstract was modified as your order instruction (justification, objectives, method, results, conclusions).
“Abstract: Previous studies mainly focused on the negative effects of teacher-student conflict, and the positive effects of conflict had rarely been mentioned. This paper suggests encouraging conflict could act as a teaching method to improve students’ innovative competence. This study has three objectives: (1) Describe the process of two kinds of teacher-student conflicts and explain the features and different situations during teacher-student interaction; (2) Exam how various types of teacher-student conflicts affect students' innovative competence; (3) Identify the mediating role of psychological safety climate in the association between conflict and students’ innovative competence. To achieve the objectives, we used evidence of 1207 university students, multivariable logistic regression analysis revealed that conflicts were associated with students’ innovative competence, and the mediation role of Psychological Safety Climate is significant. Specifically, the results revealed that Cognitive Conflict had significant positive effects on students’ innovative competence, whereas Affective Conflict had a significant negative effect on students’ innovative competence. In addition, we clarified psychological safety climate as the boundary condition for the relationship between conflict and students’ innovative competence. ”
Reviewer 2 Report
My congratulations to the authors. They have made a great effort to improve the article.
Minor changes are noted in comments file.
It is convenient to improve the Flow Chart figure.
I have allowed myself to make a proposal.
It is intended to specify how the model of structural equations organizes the theoretical variables (question 1, question 2, ...) into latent factors (F1, F2, F3, F4).

Author Response
We appreciated your careful reading of our manuscript and the constructive comments. We have thoroughly addressed these comments in the revision, incorporating the majority of recommended changes as described below in our detailed point-by-point responses. The revisions of the manuscript according to the comments are marked in red (Response). The detailed revisions in the manuscript and the responses are as follows.
Comment 1: Minor changes are noted in the comments file.
Response 1: All the problems had been solved based on your wonderful advice mentioned in the pdf file, and you can check it in the new version.
Comment 2: It is convenient to improve the Flow Chart figure.
Response 2: It has been improved by following your suggestions.
Comment 3: It is intended to specify how the model of structural equations organizes the theoretical variables (question 1, question 2,...) into latent factors (F1, F2, F3, F4).
Response 3:
The measurement of all the theoretical variables has established measurement scales already, and we used these classic scales. We used the 5-level Likert scale as the response, it is a type of psychometric response scale in which responders specify their level of agreement to a statement typically in five points. All items used the response alternatives of Strongly disagree; Disagree; Neither agree nor disagree; Agree; Strongly agree, coded into 1, 2, 3, 4, and 5, respectively.
Research question 1: How do various types of teacher-student conflicts affect students' innovative competence;
This question concludes with 2 variables (the Independent variable and Dependent variable)
Independent variable: teacher-student conflicts (Cognitive Conflict and Affective Conflict).
Cognitive conflict: It was assessed using the scale developed by Mooney [21]; a sample item is “Teacher and I frequently have disagreements about the knowledge issue”.
Affective conflict: It was assessed using the scale adopted by Mooney [21]; a sample item is “I will not make friends with my teacher”.
A greater total score corresponded to a greater level of conflict.
Dependent variable:
Students’ innovation competence: We adapted the scales by Ovbiagbonhia [1]. For example, the students were asked to reply to the item, “I am excited by unanswered questions”.
Research question 2:Identify the mediating role of psychological safety climate in the association between conflict and students’ innovative competence.
This question concludes with 3 variables (Independent variable, Mediate variable and Dependent variable)
The measurements of the Independent variable and Dependent variable are the same as the method in Question 1.
Mediate variable:
Psychological safety climate was assessed using the scale developed by Newman [47]; a sample item is “It is easy to put forward different ideas in the class”.
Reviewer 3 Report
Congratulations for such a fantastic and original work. I really liked it and I only have small suggestions for improvement, which I present below:
- The abstract is out of order. The justification for the study must be presented firstly, then the objectives, then the method, then results and finally conclusions.
- I recommend presenting the objectives and hypotheses together, and at the end of the literature review (just before the Method section). Presenting the objectives before finishing the literature review is not consistent with the scientific process, as is separating them from the hypotheses that are derived, and doing so at different times.
- Of 49 academic references cited throughout the text, only 5 of them appear in the discussion. You should cite and compare the results of the study with some more previous work with respect to those cited, either because the results contradict those previous studies or because they support them.
- Conclusions must be free of citations. All comparisons with other previous studies are made in the discussion.
Author Response
We appreciated your careful reading of our manuscript and the constructive comments. We have thoroughly addressed these comments in the revision, incorporating the majority of recommended changes as described below in our detailed point-by-point responses. The revisions of the manuscript according to the comments are marked in red (Response), and purple (Refine paragraph in the paper), respectively. The detailed revisions in the manuscript and the responses are as follows.
Comment 1:The abstract is out of order. The justification for the study must be presented firstly, then the objectives, then the method, then results, and finally conclusions.
Response 1: The abstract was modified as per your instruction.
“Abstract: Previous studies mainly focused on the negative effects of teacher-student conflict, and the positive effects of conflict had rarely been mentioned. This paper suggests encouraging conflict could act as a teaching method to improve students’ innovative competence. This study has three objectives: (1) Describe the process of two kinds of teacher-student conflicts and explain the features and different situations during teacher-student interaction; (2) Exam how various types of teacher-student conflicts affect students' innovative competence; (3) Identify the mediating role of psychological safety climate in the association between conflict and students’ innovative competence. To achieve the objectives, we used evidence of 1207 university students, multivariable logistic regression analysis revealed that conflicts were associated with students’ innovative competence, and the mediation role of Psychological Safety Climate is significant. Specifically, the results revealed that Cognitive Conflict had significant positive effects on students’ innovative competence, whereas Affective Conflict had a significant negative effect on students’ innovative competence. In addition, we clarified psychological safety climate as the boundary condition for the relationship between conflict and students’ innovative competence. ”
Comment 2:I recommend presenting the objectives and hypotheses together, and at the end of the literature review (just before the Method section). Presenting the objectives before finishing the literature review is not consistent with the scientific process, as it separates them from the hypotheses that are derived, and does so at different times.
Response2:
We presented the objectives and hypotheses together at the end of the literature review.
Comment 3: Of 49 academic references cited throughout the text, only 5 of them appear in the discussion. You should cite and compare the results of the study with some more previous work with respect to those cited, either because the results contradict those previous studies or because they support them.
Response3:
The comparison part and new academic references were added to the discussion.
“Firstly, our findings supported that cognitive conflict promoted students’ innovative competence. The literature proved that cognitive conflict had positive outcomes (such as group performance) in companies and top management teams [16, 18-22]. We found that cognitive conflict between teachers and students had a beneficial impact as well.
Secondly, consistent with the studies of negative outcomes of affective conflict [3,7, 36, 37], the results indicated that affective conflict was negatively correlated with students’ innovative competence.
Thirdly, we found that a psychological safety climate strengthened the advantages of cognitive conflict and weakened the disadvantages of affective conflict, which proved the third goal. Previous studies had found that psychological safety climate promotes knowledge sharing, learning behavior, and team creativity [28, 30, 31].”
Comment 4:Conclusions must be free of citations. All comparisons with other previous studies are made in the discussion.
Response 4: The sentence which has citations was deleted in the conclusion, and all comparisons are made in the discussion.
This manuscript is a resubmission of an earlier submission. The following is a list of the peer review reports and author responses from that submission.
Round 1
Reviewer 1 Report
Thanks for the invitation to review this manuscript. I carefully read through this study and find many critical problems shown as below.
(1) The measure of "students’ innovation competence" was adapted from a scales by Ovbiagbonhia et al. (2019). However, such adaption lacks theoratical support and without necessary explanation. In Ovbiagbonhia et al. (2019), six aspects of competence were included (i.e., creativity, energy, leadership, creative self-efficacy, risk propensity, solving ambiguous problems). In contrast, only three items were selected (i.e., leadership, solving ambiguous, risk tolerance). The scale has been truncated and thus is incorrectly applied. Moreover, the phrase of items is also changed. For example, In Ovbiagbonhia et al. (2019), the item of “I am challenged by unanswered questions" is changed to be “I am excited by unanswered questions" in this manuscript. This article has neither shown any rationality of the adaption nor results of exploratory factor analysis (EFA).
(2) The correlation among variables are highly inter-correlated. For example, in Table 1, the correlation between CC and G2 is as high as 0.663. The inclusion of highly correlated variables would inevitably induce multi-colinearity and biased estimation.
(3) The results of CFA, Chronba Alpha, AVE, CR are all absent. The validity of measure is doubtful.
(4) The development of hypothses lacks theoratical support, and is far from being founded.
Author Response
We appreciated your careful reading of our manuscript and the constructive comments. We have thoroughly addressed these comments in the revision, incorporating the majority of recommended changes as described below in our detailed point-by-point responses. The revisions of the manuscript according to the comments are marked in red (Response). The detailed revisions in the manuscript and the responses are as follows.
Point1:The measure of "students’ innovation competence" was adapted from a scale by Ovbiagbonhia et al. (2019). However, such adaption lacks theoratical support and without necessary explanation. In Ovbiagbonhia et al. (2019), six aspects of competence were included (i.e., creativity, energy, leadership, creative self-efficacy, risk propensity, solving ambiguous problems). In contrast, only three items were selected (i.e., leadership, solving ambiguous, risk tolerance). The scale has been truncated and thus is incorrectly applied. Moreover, the phrase of items is also changed. For example, In Ovbiagbonhia et al. (2019), the item of “I am challenged by unanswered questions" is changed to be “I am excited by unanswered questions" in this manuscript. This article has neither shown any rationality of the adaption nor results of exploratory factor analysis (EFA).
Response 1: The measurement of students’ innovative competence combined six aspects (creativity, energy, leadership, creative self-efficacy, risk propensity, solving ambiguous problems) in Ovbiagbonhia et al. (2019). These six features were derived from three separate authors, Ovbiagbonhia et al. (2019) integrated them with good validation.
We don't use the terms "creativity" and "creative self-efficacy" because they're always used to assess pupils' creativity. Ovbiagbonhia et al. (2019) conclude creativity in their innovation measurement, while we separate these two concepts in our paper. For the “energy” section, Cronbach's alpha was 0.375, hence, we decided to remove it.
As a result, this paper finally chose three of them, which have been proven with excellent validation (See Table 3 attached for more information on Factor Loading, Cronbach's alpha, CR, AVE, and VIF).
The rationale for the change in expression from "I am challenged by unresolved questions" to "I am excited by unanswered questions" is because after the Pivot survey received a high Cronbach's alpha, we conducted in-depth interviews to find out why.
Students are perplexed by the original item, which has a phrase that is similar to one of the items in "Cognitive Conflict." We gathered input from interviewees and discussed it with five academics before changing the item and increasing the Cronbach's alpha to 0.858.
Point 2: The correlation among variables are highly inter-correlated. For example, in Table 1, the correlation between CC and G2 is as high as 0.663. The inclusion of highly correlated variables would inevitably induce multi-colinearity and biased estimation.
Response 2: The highly inter-correlated variables are definitely a problem. However, the following characteristics of students' innovation competence are positive (Chronba Alpha = 0.858, AVE = 0.861, CR = 0.771, VIF=1.822), indicating that the data is still valid in the next steps.
Point 3: The results of CFA, Chronba Alpha, AVE, CR are all absent. The validity of measure is doubtful.
Response 3: Please accept my apologies for the missed explanation, specifics can be found in the attached Table3.
Point 4: The development of hypothses lacks theoratical support, and is far from being founded.
Response 4: Hypotheses 1 and 2 are based on conflict theory, which originated from team management, this paper explores the connotation is transformed in the classroom. Regarding Hypotheses 3 and 4, we contributed to the mediating role of psychological safety climate based theory in educational psychology. These theories were detailed in the literature section.

Reviewer 2 Report
Dear authors, I consider that the presented work is interesting, novel and well developed. However, I have some suggestions that can strengthen it:
- You can include the research objectives in the abstract, since they are not reflected.
- Regarding the above, the abstract states that the work "proposes an innovative teaching method". It is recommended to review whether it is convenient to refer to a teaching method or a pedagogical topic, since they are different.
- The article does not explain what an innovative pedagogy is, when it is a key word of the study.
- Order the exposition of the topics in the introduction, to facilitate reading; for example, it begins with conflict, then innovation competition is addressed, then it returns to conflict.
- It is recommended to make precisions in the materials and methods section: what was the temporality of the study? What characteristics do the American universities that you included in the study have? What specialization are the authors referring to when describing the selection of students? What were the response options in the items that measured the selected variables?
- In the results section, the creativity of the students is described, when it had not been mentioned previously. It is suggested to clarify.
Author Response
We appreciated your careful reading of our manuscript and the constructive comments. We have thoroughly addressed these comments in the revision, incorporating the majority of recommended changes as described below in our detailed point-by-point responses. The revisions of the manuscript according to the comments are marked in red (Response), and purple (Refine paragraph in paper), respectively. The detailed revisions in the manuscript and the responses are as follows.
Point1: You can include the research objectives in the abstract, since they are not reflected.
Response 1: At the end of the abstract, we added the 3 objectives.
“This study has three objectives: (1) Describe the process of two kinds of teacher-student conflicts and explain the features and different situations during teacher-student interaction; (2) Exam how various types of teacher-student conflicts affect students' innovative competence; (3) Identify the mediating role of psychological safety climate in the association between conflict and students’ innovative competence.”
Point2: Regarding the above, the abstract states that the work "proposes an innovative teaching method". It is recommended to review whether it is convenient to refer to a teaching method or a pedagogical topic, since they are different. The article does not explain what an innovative pedagogy is, when it is a keyword of the study.
Response 2: The expression of “innovative pedagogy” is not appropriate, we change the words “teaching method” in the abstract. Because this paper strengthened the teaching method which encourages cognitive conflict in the teacher-student interaction and is not an innovative pedagogy. We also removed "innovative pedagogy" from the keyword list because it was only referenced a few times in the paper.
Point3: Order the exposition of the topics in the introduction was changed as your wonderful suggestions and the key points in each paragraph were listed as follows.
Response3:
Para1: Students’ innovation competence.
Para2-4: Conflict
Para4: Psychological safety climate
Para5: Objectives
The refined Introduction section was attached.
Point4: It is recommended to make precisions in the materials and methods section: what was the temporality of the study? What characteristics do the American universities that you included in the study have? What specialization are the authors referring to when describing the selection of students? What were the response options in the items that measured the selected variables?
Resopnse4: Please accept my apologies for the missed explanation, specifics can be found in the attachment.
Point5: In the results section, the creativity of the students is described, when it had not been mentioned previously. It is suggested to clarify.
Response5: Students' creativity should be changed to students' innovation competency in the results section. And the two incorrect expressions were modified as well.

Reviewer 3 Report
The structure of the article needs to be revised and the content needs to be clarified.
- INTRODUCTION.
The introduction needs to be revised. The information included is adequate, although it needs to follow a structure that responds to the presentation of general information (the concepts and definitions that the authors include) where information is lacking regarding the innovative competence of the students, previous research (literature review), it would be convenient to complete with previous studies, if any, on the relationship of cognitive conflict with the innovative competence of the student and on the relationship of affective conflict with the innovative competence of the student.
The working hypotheses should be specified in the aims or objectives of the work.
- MATERIALS AND METHODS
2.1. Participants
What information is accessed and how?
-Questionnaires administered by teachers and volunteers in their classrooms.
-Questionnaires completed in electronic format and sent by email.
-In-depth interviews
CONCRETE validity/reliability of questionnaires. Description of participants for each of the data collection procedures.
It would be interesting to have a flow chart in which it would be possible to know the sequence followed.
IT WOULD BE INTERESTING TO HAVE A SECTION FOR THE DESCRIPTION OF THE INSTRUMENT AND TO CLARIFY THE DIMENSIONS OF EACH VARIABLE TO BE ASSESSED.
Three scales are used:
- Measurement of students' innovation competence:
The scales of Ovbigbonnia et al., 2019 are ADAPTED.
- The scale of Mooney et al., 2007, to measure cognitive conflict and affective conflict.
- The scale of Newman et al., 2017, to measure psychological safety climate.
2.2. Measurements
WHAT AM I GOING TO MEASURE AND HOW?
It is necessary to explicitly state which variables I will measure and how I will measure them. The comparisons I am going to make, the techniques and tools I am going to use.
Organise this information on the basis of the structure of the hypotheses/objectives of the work.
- RESULTS
Organise the presentation of the results based on the structure of the hypotheses/objectives of the work.
- DISCUSSION
Organise the presentation of the discussion based on the structure of the working hypotheses/objectives.
- CONCLUSIONS
Organise the conclusions based on the structure of the working hypotheses/objectives.
Limitations and suggestions for future research

Author Response
We appreciated your careful reading of our manuscript and the constructive comments. We have thoroughly addressed these comments in the revision, incorporating the majority of recommended changes as described below in our detailed point-by-point responses. The revisions of the manuscript according to the comments are marked in red (Response), and purple (Refine paragraph in the paper), respectively. The detailed revisions in the manuscript and the responses are as follows.
Point 1:
The introduction needs to be revised. The information included is adequate, although it needs to follow a structure that responds to the presentation of general information (the concepts and definitions that the authors include) where information is lacking regarding the innovative competence of the students, previous research (literature review), it would be convenient to complete with previous studies, if any, on the relationship of cognitive conflict with the innovative competence of the student and on the relationship of affective conflict with the innovative competence of the student.
Response 1:
Order the exposition of the topics in the introduction was changed to your wonderful suggestions and the key points in each paragraph were listed as follows.
Para1: Students’ innovation competence.
Para2-4: Conflict
Para4: Psychological safety climate
Para5: Objectives
The refined Introduction section is in the attachment.
Point 2: The working hypotheses should be specified in the aims or objectives of the work.
Response 2: new paragraph was added at the beginning of Hypotheses Part
“In order to achieve objective 2 (Examine how various types of teacher-student conflicts affect students' innovative competence), H1 and H2 were demonstrated based on conflict theory. For objective 3 (Identify the mediating role of psychological safety climate in the association between conflict and students’ innovative competence), H3 and H4 were demonstrated based on the previous literature about psychological safety climate.”
Point 3:
2.1. Participants
What information is accessed and how?
2.2. Measurements
WHAT AM I GOING TO MEASURE AND HOW?
It is necessary to explicitly state which variables I will measure and how I will measure them. The comparisons I am going to make, the techniques and tools I am going to use.
Organise this information on the basis of the structure of the hypotheses/objectives of the work.
Response 3:
The methodology section was modified as your inspired suggestion. The details are in the attachment.
Point 4:
RESULTS
Organise the presentation of the results based on the structure of the hypotheses/objectives of the work.
Response 4:
We added more content to specify how the hypotheses and objectives was supported.
“The results of the hierarchical regression shown in Table 5 (M4) support Hypotheses 1 and 2, which achieved the objective 2. Cognitive conflict had significant positive effects on students’ innovative competence (β = 0.319, p < 0.01). Affective conflict was significantly negatively related to students’ innovative competence (β = -0.505, p < 0.01).
As shown in Table 5 (M2), cognitive conflict had significant positive effects on PSC (β = 0.275, p < 0.01), and affective conflict had a significant negative effect on PSC (β = -0.209, p < 0.01). As shown in Table 5 (M5), PSC was significantly and positively correlated with students’ innovative competence (β = 0.227, p < 0.01). These patterns are consistent with Hypotheses 3 and 4, which resulted in the achievement of Objective 3.”
Point 5:
DISCUSSION
Organise the presentation of the discussion based on the structure of the working hypotheses/objectives.
Response 5:
We added more content to specify how the hypotheses and objectives was supported.
“Firstly, our findings supported that cognitive conflict promoted students’ innovative competence. Cognitive conflict facilitates the surfacing of different ideas and viewpoints. It helps students reconsider the knowledge from various angles. In contrast to students’ silence or blind acceptance of knowledge, cognitive conflict increases learning feedback and in more extensive debates which results in new understanding and more perspectives.
Secondly, consistent with the studies of negative outcomes of affective conflict [3, 36, 37], the results indicated that affective conflict was negatively correlated with students’ innovative competence. Affective conflict results in a discussion that is off-topic for class content. And it evokes negative emotions and amplifies aggression. And these findings achieved the first and second goals mentioned early.
Thirdly, we found that a psychological safety climate strengthened the advantages of cognitive conflict and weakened the disadvantages of affective conflict. And it proved the third goal. ”

Round 2
Reviewer 1 Report
The authors have clarified a few problems. However, the current revised version is still far from the basic standard of publication.
(1) The results of CFI (as claimed by authors to have supplemented in table 3) is rather incomplete. It must contain CFI, TLI, RMSEA, SRMR, Chi2/df.
(2) The results of exploratory factor analysis (EFA) could not be omited, as the phrase and items has been changed and is inconsistent with original scale. The current revision has not well addressed this issue.
(3) The control variables only include "gender" and "grade". It is easy to expected that academic performance, test score, personalities can affect students' innovative behavior. The current control variables seem to omit many factors that should be considered.
(4) A most important point is the development of hypotheses. The development of hypotheses is rather rough and lacks corresponding theoratical support. In the author response, the authors argue that the conflict theory and psychological safety climate based theory in educational psychology are used. Unfortunately, I have not seem sound theoratical deduction in the hypothetical development. I suggest the authors refering to literature in the field of educational psychology and dealing with the development of hypotheses seriously.
(5) Results of table 4 obviously contain many errors. As shown by authors, a correlation value 0.269 has been labeled as p<0.01, however, some much greater values (e.g., 0.797, 0.663, 0.568, 0.576) are lebeled as statistically insignificant (no asterisk). I feel sorry to see such mistakes in a serious discussion.
Reviewer 2 Report
Dear authors,
You are presenting a better version. I can read the article more fluently; however, in the methodology I was unable to identify the timing of the study. When was it carried out? Could you briefly specify? Please.
Success with the publication.
Kind regards.
Reviewer 3 Report
Reference is made to the use of a validity test for the questionnaire, but in reality an analysis of reliability or internal consistency is carried out.
Page 5 of 14, line 211. It indicates "while the lesser validation items (Cronbach's alpha)", it could indicate "while the lesser RELIABILITY items (Cronbach's Alpha<0.3)
Table 3. Internal consistency analysis of the questionnaire.
Revise wording and reconsider need for bootstrapping (page 6 of 14, lines 262 to 265):
Finally, to further test the mediating effect (H3, H4), we estimate the indirect, direct, and total effects as well as their 95% bias-corrected confidence intervals (CI), the 5,000- replication bootstrapping procedure was used based on the PROCESS macro adapted from Hayes. And H3 and H4 were also supported.
The use of the bootstrapping technique does not seem necessary given that the sample of 1183 informants.